# Advancing Newborn Screening Long-Term Follow-Up: Integration of *Epic*-Based Registries, Dashboards, and Efficient Workflows

**DOI:** 10.3390/ijns10020027

**Published:** 2024-03-25

**Authors:** Katherine Raboin, Debra Ellis, Ginger Nichols, Marcia Hughes, Michael Brimacombe, Karen Rubin

**Affiliations:** 1Connecticut Newborn Screening Network, Connecticut Children’s, Hartford, CT 06106, USA; kraboin@connecticutchildrens.org (K.R.); dellis@connecticutchildrens.org (D.E.); gnichols01@connecticutchildrens.org (G.N.); 2Center for Social Research, University of Hartford, Hartford, CT 06105, USA; mhughes@hartford.edu; 3Research Operations and Development, Connecticut Children’s, Hartford, CT 06106, USA; mbrimacombe@connecticutchildrens.org; 4Department of Pediatrics, University of Connecticut School of Medicine, Hartford, CT 06106, USA

**Keywords:** newborn screening, long-term follow-up, *Epic* registries, dashboards, key performance metrics, workflows, care coordination, continuous improvement, population health, nurse analyst

## Abstract

The Connecticut Newborn Screening (NBS) Network, in partnership with the Connecticut Department of Public Health, strategically utilized the *Epic* electronic health record (EHR) system to establish registries for tracking long-term follow-up (LTFU) of NBS patients. After launching the LTFU registry in 2019, the Network obtained funding from the Health Resources and Services Administration to address the slow adoption by specialty care teams. An LTFU model was implemented in the three highest-volume specialty care teams at Connecticut Children’s, involving an early childhood cohort diagnosed with an NBS-identified disorder since the formation of the Network in March 2019. This cohort grew from 87 to 115 over the two-year project. Methods included optimizing registries, capturing external data from Health Information Exchanges, incorporating evidence-based guidelines, and conducting qualitative and quantitative evaluations. The early childhood cohort demonstrated significant and sustainable improvements in the percentage of visits up-to-date (%UTD) compared to the non-intervention legacy cohort of patients diagnosed with an NBS disorder before the formation of the Network. Positive trends in the early childhood cohort, including %UTD for visits and condition-specific performance metrics, were observed. The qualitative evaluation highlighted the achievability of practice behavior changes for specialty care teams through responsive support from the nurse analyst. The Network’s model serves as a use case for applying and achieving the adoption of population health tools within an EHR system to track care delivery and quickly fill identified care gaps, with the aim of improving long-term health for NBS patients.

## 1. Introduction

The Connecticut Newborn Screening Network (the Network) is a statewide network funded through and in direct partnership with the Connecticut Department of Public Health. Established at Connecticut Children’s in 2018 in collaboration with Yale New Haven Hospital, the Network responds to all reports of infants with an out-of-range newborn screen (NBS). In coordination with the infant’s primary care provider (PCP), the Network begins the diagnostic work-up and supports the family through pre-diagnosis. If an infant is diagnosed with a disorder, the Network coordinates initial treatment and long-term follow-up (LTFU) care with the appropriate specialty care team for the condition identified, working with hospital-based providers, PCPs, and specialists statewide.

The Network has strategically harnessed the capabilities of the electronic health record (EHR) system to establish and manage clinical registries for short and long-term follow-up of NBS populations. Clinical registries within the EHR system are emerging as powerful population health management tools for healthcare improvement in targeted populations of patients [1,2]. This integrated approach allows for real-time data collection and display through hierarchical design, including an overall Network/program-level registry and disease-specific subspecialty registries. For this project, the Network used its existing EHR clinical documentation, health information exchange, and population health modules: EpicCare Ambulatory, Care Everywhere, and Healthy Planet, trademarks of *Epic* Systems Corporation. The *Epic* registry defines populations utilizing inclusion rules, making reporting, risk scoring, and performance tracking efficient. Complementing the registry, the Network employs additional tools such as performance metrics, dashboards, and health maintenance to extract maximum value. The dynamic visualizations on *Epic* dashboards, featuring key performance metrics (KPMs), offer a holistic view of performance across various dimensions. The interactive nature of these dashboards allows providers to easily access patient-level data. This ability empowers providers and the Network program coordinators to track progress and quickly fill care gaps, which contributes to improved patient outcomes.

Soon after building and launching the NBS LTFU registry and associated dashboards, the Network faced slow adoption by specialty care teams due to the lack of desired registry optimization and the provision of ongoing training and support in using the registry dashboards effectively. The Network successfully garnered the resources needed to address these challenges by securing an NBS LTFU grant from the Health Resources and Services Administration (HRSA). This award allowed the Network to refine registry content, advance data collection facilitated by *Epic*-based tools, streamline processes and workflows, support a nurse analyst position, and advance tool usage by specialty care teams. These enhancements addressed existing cases and ensured the seamless incorporation of new patients into the system.

The Network’s LTFU model has introduced and evaluated innovations and emerging technology not previously described in the literature. These innovations effectively address barriers commonly cited by previous studies, such as limited resources and a lack of access to reliable clinical patient data [3,4]. In addition, the Network model exemplifies the four components of LTFU care proposed by the Advisory Committee on Heritable Disorders in Newborns and Children (ACHDNC): care coordination, evidence-based practice, continuous improvement, and new knowledge discovery [5,6].

## 2. Materials and Methods

### 2.1. Registry Optimization

As described in the introduction, the clinical registries exist within the EHR system. The Network began implementing its LTFU approach with specialists from the three NBS specialty care teams at Connecticut Children’s with the highest volumes of confirmed NBS patients—endocrinology, genetics, and hematology. In close collaboration with each specialty care team, essential optimizations of the disorder-specific subregistries and associated population health tools were implemented. This effort involved identifying additional data points from patient records, including specific comorbidities, medications, procedures, encounters, lab results, and referrals, to input into the registries and reports. The Network registry is programmed to collect these data automatically and display performance metrics on dashboards. The registry collects over 200 data points on different comorbidities, procedures, labs, medications, surgery dates, ED and hospitalizations, growth measures, referrals, and encounter dates with specialists and PCPs.

The continually growing early childhood cohort tracked by the registry comprises 322 patients from birth to four years with a confirmed diagnosis through the Network since 1 March 2019. The cohort growth rate is 60–70 patients a year since new patients are added to the LTFU registry once they are confirmed for a disorder. Patients who move out of state are removed from the LTFU registry, and their medical records are sent to the receiving care team. Deceased patients are also removed from the registry. For this project, only a subset of this cohort who receives LTFU care by one of the above-mentioned care teams at Connecticut Children’s was tracked (*n* = 87 at start and *n* = 115 at finish).

### 2.2. Capturing External Data from Health Information Exchanges

The Network performed extensive analysis and enhancements to optimize the registry’s seamless incorporation of external data. An advantage of utilizing *Epic* registries is the ability to retrieve clinical data from external sources through Care Everywhere, *Epic*’s health information exchange. This functionality empowers the registry to gather information from external sites like primary care practices, imaging facilities, and hospitals. This enhanced interoperability results in a more unified representation of the patient’s long-term care journey, offering a comprehensive and less fragmented view across various healthcare settings. This alleviates the burden of manual data entry and reduces the associated human error.

### 2.3. Incorporating Evidence-Based Guidelines and Decision Support

The overall design of the registries and related population health tools was aligned with evidence- or expert consensus-based guidelines, NBS frameworks [7], and insights from published articles on *Epic*-based registries [8]. A vital part of the project involved integrating clinical decision support directly into *Epic* in the form of health maintenance. Health maintenance is *Epic*’s primary tool for monitoring and addressing gaps in patient care. Within this framework, alerts can be configured to assist clinicians in tracking essential labs, visits, or procedures required for managing patients with chronic diseases. Health maintenance is prominently highlighted in red if a care gap exists at the point of care, alerting the provider to upcoming care requirements and facilitating timely interventions, visualization of the percentage of visits up-to-date (%UTD), and adherence to best practices.

### 2.4. Selection of Quality Measures and Dashboards

The registry dashboards display the percentage of patients up-to-date on key components of care tracked as a health maintenance topic. This allows care teams to identify patients that are overdue for care quickly. These percentages serve as KPMs, assessing overall care effectiveness. Further supporting clinician needs, the dashboards feature links to additional disorder-specific reports, such as lists of patients on specific medication, those of specific age(s), or those overdue for a critical care component. The dashboard is displayed upon logging into the EHR system. This information empowers care teams to quickly and proactively connect with patients to bridge care gaps and ensure timely interventions. The Network’s registry design and the dashboards enable efficient monitoring, reporting, and decision making at every level. A Network program-level dashboard was created to view patients in the entire cohort, and each specialty care team has a specialty-specific dashboard showing KPMs relevant to their population specialty. Figure 1 shows an example of the Network’s program-level dashboard, while Figure 2 displays an partial view of a specialty care team dashboard: the hematology care team dashboard.

### 2.5. Attention to Workflows and Training

In addition to the significant efforts invested in optimizing registries and associated population health tools, substantial attention was dedicated to enhancing workflows and providing training on using the dashboards and health maintenance tools. New tasks were allocated to the most appropriate and available staff for workflow development. Task assignments varied for each specialty care team. For example, in genetics, the genetic counselor was responsible for reviewing the dashboard weekly to identify patients overdue for a visit. The genetic counselor would then notify the medical assistant, who would contact the family to schedule a visit. In endocrinology, the nursing staff reviewed the dashboard and contacted the family to schedule an appointment. In hematology, a nurse practitioner reviewed the dashboard and alerted the medical assistants to schedule a visit. At the start of the project, dashboard reviews for a department could take 30 min to an hour, as there were many patients overdue for visits. Because it was time-consuming, staff tended to review the dashboards weekly. However, as the project progressed, fewer patients were overdue, dashboard reviews became quicker, and staff could conduct daily reviews. The Network’s nurse analyst collaborated closely with specialty care teams, contributing to the creation of the most feasible and streamlined workflows. The three specialty care teams received comprehensive training on utilizing the dashboards to identify patients approaching due dates or overdue for care.

The Network and specialty care teams conducted quarterly Zoom meetings throughout the two-year project, fostering an environment of trust and support and increasing engagement. This meeting interval was determined based on feedback from care teams. The nurse analyst was available to support the specialty care teams continuously between the structured meetings, which provided confidence to the staff to use their dashboards independently. During the meetings, the nurse analyst presented the improvements in KPMs and celebrated the specialty care team’s progress and success. In addition, cases with care gaps were reviewed to gain insights for process improvement. For example, appropriate referrals to care coordination were initiated for patients exhibiting frequently missed appointments or a lack of follow-through on an urgent care interventions.

### 2.6. Quantitative Methods

Trends in %UTD on Visits: Visit timeliness is a primary driver of improvements in other KPMs since the patient management plan is typically co-developed between provider and family, and related orders are placed at the visit. Therefore, the main KPM that was tracked was %UTD on visits. Each care team determined the frequency of visits based on patient age and condition. Trends in %UTD on visits in each early childhood cohort in endocrinology, genetics, and hematology were monitored on the registry dashboard quarterly between August 2021 and August 2023.

Comparison of %UTD on Visits Between Early Childhood Intervention and Legacy Cohort: The Network aimed to assess the statistical significance of observed improvements in visit timeliness between these two cohorts. The early childhood intervention cohort comprised patients from birth to four years, diagnosed since the formation of the Network in March 2019, and followed by one of the three specialty care teams (*n* = 87 at start and *n* = 115 at finish), and represented the cohort to which the LTFU intervention was applied. In contrast, the legacy non-intervention cohort included patients older than four years, diagnosed through NBS before March 2019, and followed by the respective specialty care teams (*n* = 612).

Trends in Additional KPMs: The three specialty care teams prioritized additional KPMs that were most important for their specialty. Examples include the percentage of patients with SCD up-to-date on transcranial doppler and patients with congenital hypothyroid up-to-date on labs. Those were also reviewed quarterly by the care teams for improvements.

### 2.7. Quantitative Analysis Methods

De-identified data for statistical analysis was abstracted from the registry and sent to a biostatistician.

Trends in %UTD on Visits: The %UTD on visits was compared in six-month intervals using chi-square tests for endocrine, genetics, and hematology patients.

Comparison of %UTD on Visits Between Early Childhood Intervention and Legacy Cohort: To determine the statistical significance of results with a sufficient *n*, comparisons were performed between the %UTD on visits in the early childhood intervention cohort and a legacy non-intervention cohort. A time series plot of monthly counts broken down by NBS and legacy groups was obtained for each variable of interest. Overall median values are shown for each group as a reference. Assuming the monthly counts to be independent, two-sample Mann–Whitney non-parametric tests were conducted to detect significant median differences between NBS and legacy groups. In addition, time series linear trend analysis for monthly counts was conducted to obtain 12-month forecasts for several variables. *p*-values < 0.05 were reported as significant.

Trends in Additional KPMs: Most measures had too small a sample size for statistical analysis. However, the percentage of patients with congenital hypothyroid up-to-date on TSH and Free T4 was also compared quarterly using chi-square tests.

### 2.8. Qualitative Methods

The Network employed an external program evaluator to interview and conduct focus groups with the genetics, endocrine, and hematology care teams over the project period. The focus groups occurred during the teams’ regularly held meetings to review dashboard information. All members of the care teams participated in at least one of the focus groups. However, the majority attended both, including subspecialty care team doctors, nurses, administrative support staff, and other clinical staff, including the dietician and the genetic counselor on the genetics care team and the clinical social worker on the hematology team. In addition, the program evaluator periodically attended care team reviews of dashboard information to better understand their review process and use this understanding to facilitate focus group discussions. Interview questions were designed to investigate how the introduction of population health tools, particularly the routine monitoring of NBS registry dashboards, was experienced. The analysis focused on whether or not the staff and their patients benefited from using registry tools and nurse analyst support to elucidate workflow challenges and ideas for continuous improvement moving forward.

## 3. Results

### 3.1. Trends in %UTD on Visits

Improvement was observed in the percentage of patients who were up-to-date on visits in all specialty care teams (Figure 3). The %UTD on visits for endocrine, genetics, and hematology patients increased from 67% to 90%, 60% to 85%, and 79% to 97%, respectively.

Chi-square tests for equality of proportions were conducted to compare six-month intervals—baseline (August-21), six months (Februrary-22), twelve months (August-22), eighteen months (February-23), and two years (August-23). Significant differences in endocrinology were found between baseline and twelve months (adjusted *p*-value = 0.009), between baseline and eighteen months (adjusted *p*-value = 0.029), and between baseline and two years (adjusted *p*-value = 0.011). Significant differences in genetics and hematology were not found.

### 3.2. Comparison of %UTD on Visits between Early Childhood and Legacy Cohort

Time series plots for each specialty are shown in Figure 4, Figure 5 and Figure 6 and indicate that the improvements in the % of patients UTD on visits were significant for each specialty area. Mann–Whitney non-parametric comparison of median differences (assuming that the monthly counts are independent) reached statistical significance (*p*-value < 0.001). Figure 7 displays the improvements in the cohort aggregate (early childhood cohort compared to legacy cohort patients). A time series-based trend analysis for the difference between early childhood and legacy patients provides a 12-month forecast in Figure 8, showing that improvements will likely continue to improve over time.

### 3.3. Trends in Additional KPMs

Each specialty care team pre-selected additional KPMs to track during the two-year project period (Table 1). Positive trends were observed in endocrinology, genetics, and hematology KPMs for their congenital hypothyroidism (CH), very long-chain acyl-CoA dehydrogenase deficiency (VLCADD), and sickle cell disease (SCD) cohorts, respectively.

For Table 1, chi-square tests for equality of proportions were conducted to compare six-month intervals for endocrinology (row 1). No significant differences were observed. Sample sizes in other categories were too small for formal significance testing.

### 3.4. Qualitative Results

Analyses of feedback from all three specialty care teams were aggregated to provide an overview of their experiences of working to incorporate the registry and dashboards into their workflows. Three thematic areas rose to the surface: Workflow Logistics and Roles, High-Risk Cases, and Perceived Value and Future Uses of the Dashboard.

Workflow Logistics and Roles: The development of workflow logistics and personnel responsible for utilizing the registry and dashboards evolved over the project period among all specialty care teams. Workflows were understandably different given the different specialty care team contexts, with some team members embracing the dashboards more readily than others. Direct care providers were sometimes less receptive to adding dashboard review to their existing care burden. However, they valued utilizing staff who could regularly review the dashboards and respond to care gaps, with escalation if necessary. Specialty care teams affirmed that adopting population health tools enhances efficiency and provides peace of mind. The variations in developing workflow logistics underscore the importance of co-developing unique workflows with each care team, as staff composition and specific clinic challenges differ.

Specialty care teams participated in quarterly metrics reviews, demonstrating a commitment to data-driven insights. Teams collaboratively raised questions about specific metrics, utilizing the nurse analyst’s expertise for adjustments to optimize data interpretation. Across all three divisions, the specialty care teams expressed deep appreciation for the support provided by the nurse analyst, highlighting in their discussion the essential role of an embedded nurse analyst for the successful optimization of dashboards. The nurse analyst’s ability to understand the care team’s health focus, address questions about technical challenges, and adjust dashboards quickly helped to establish credibility in the data reports and facilitate team uptake. The nurse analyst’s highly responsive and concrete support, a theme in each focus group discussion, became the lynchpin for the initiative’s success.

High-Risk Cases: High-risk cases, particularly those at risk of being lost to follow-up, were a central focus during team reviews. Teams engaged in problem solving, which underscored the need for care coordination services. Dashboard reviews with core members of the specialty care teams allowed for an in-depth discussion on what might be barriers to care and how to generate a follow-up plan. All three specialty care teams repeatedly discussed a gap in service roles such as care coordination, social work, and genetic counseling for addressing this population’s social, behavioral, and mental health needs, not just for families identified as high-risk cases but also for all families in this NBS population. The spotlight on high-risk cases reflects the teams’ commitment to ensuring comprehensive and compassionate patient care and the need for tools and care coordination support to address challenges identified for families to reduce barriers to treatment adherence.

Perceived Value and Future Uses of the Dashboard: As already indicated in the above sections, the feedback gathered from the focus group discussions suggests that the specialty care teams highly valued the dashboard data; moreover, when observed during dashboard reviews, all specialty care teams were actively engaged in monitoring the dashboards, contacting patients, and exploring reasons for care gaps. Dashboards were credited with tangible efficiency improvements and streamlining tasks, ultimately saving time. Dashboards also provided specialty care teams with peace of mind. As one clinician described, “You always feel like you are chasing patients off the top of your head and losing sight of those you haven’t seen for years…I like that the dashboard can identify who needs to come in at a glance, and I don’t have to search through records and charts”. Another clinician added, “It is really helpful. For years, we had to physically go through the list, which is very time-consuming; the dashboards have been helpful in care coordination…it decreases time in searching through charts and lists…or checking multiple data systems or making multiple phone calls”.

Furthermore, specialty care teams have envisioned future applications, including tracking medications and aligning metrics with U.S. News and World Report, highlighting dashboards’ evolving potential and versatility in healthcare settings. All specialty care teams showed interest in expanding dashboard displays beyond the early childhood cohort to include their legacy patients. One provider stated, “Would desperately love to do this with patients with other diagnoses…so appreciative of the NBS registry and dashboard. It helped these families…It highlights how problematic [inefficient and ineffective] it is to not have these tools for other conditions and patients”. Additionally, each specialty care team recognized how the registry and related population health tools can be leveraged to support future quality improvement projects. The program evaluator’s qualitative analysis outlined the nuanced progress and impact of dashboard utilization across the specialty care teams, shedding light on successes and areas for potential refinement in the ongoing integration of these tools.

## 4. Discussion

The Network’s model encapsulates the four core components of NBS LTFU while providing a comprehensive solution to common barriers for NBS LTFU. The model represents an ongoing effort to improve the efficiency, effectiveness, and quality of services to improve the health of the NBS population.

### 4.1. Four Core Components

Care Coordination—The specialty care teams expressed that many of their NBS patients and families have unmet care coordination needs that result in missed appointments. The clinic-level monitoring of registry dashboards assists in identifying patients with care gaps. This allowed for further conversations with families to help understand their root concerns such as a lack of transportation, food insecurity, or behavioral health needs. Providers could then refer families to care coordinators who provide services that link families to community-based resources.

Evidence-Based Treatment—It is essential that newborns diagnosed with a disorder through NBS receive care aligned with evidence- or consensus-based practice to ensure the best possible LTFU outcomes. At the onset of building the registry, specialty care teams were asked to choose KPMs that would enable them to follow existing clinical practice guidelines or best practices more reliably for their patients. The selection of KPMs for each condition-specific subregistry required collaboration and clarification by each specialty care team early on in the project. It took substantial time and effort by the nurse analyst during the initial registry building and the ongoing streamlining. As a result, the Network’s LTFU registry effectively tracks patients and is a meaningful tool, enabling providers to deliver care according to the best available evidence.

Continuous Improvement—Previous research findings underscore that relying solely on dashboards may not consistently deliver anticipated results [9,10], emphasizing the necessity of additional support for a comprehensive and impactful healthcare strategy. Cognizant of this, the Network realized early on that merely creating registry dashboards is unlikely to achieve the improved outcomes documented here. Having a nurse analyst embedded in the Network team was crucial. The nurse analyst continuously created new registries or streamlined existing ones and provided training. The literature on continuous improvement supports the idea that it is essential to engage with healthcare providers during the initial planning stages of the registry and dashboard development and throughout multiple iterative cycles [11,12]. From the onset, the Network set up a mutually agreed-upon schedule of quarterly meetings to review data trends, address emerging challenges, brainstorm innovative workflows to expedite the closure of care gaps, and refine how metrics are presented on dashboards. The nurse analyst continuously worked with the specialty care teams between meetings, conducting multiple sessions to guide them through training and ensure that the dashboards were optimized for each specialty.

These continuous improvement activities empowered specialty providers by involving them in decision making at every step of the transformation and ensuring alignment with best practices for each specialty. The nurse analyst highlighted the benefits to the specialty care teams and their patients of embracing the dashboard population health tools by reviewing the improvements in their KPMs. In addition, the nurse analyst’s availability for training helped guide specialty care teams toward a cultural shift from individual patient care to embracing a broader focus on improving outcomes for a population of patients. Over time, the specialty care teams valued their newfound ability to practice in a more proactive/preventive manner rather than in reactive mode by promptly identifying patients who missed elements of care. They also recognized decreased time and effort in tracking and identifying these high-risk patients.

New Knowledge Discovery—The importance of big data collected from multiple sites for new knowledge generation, improved clinical care, and public health surveillance is well established [13]. Developing the capacity for big data analytics for the NBS population is a shared national goal. The LTFU outcome data generated from the Network registry can be aggregated, de-identified, and exported to the evolving national NBS LTFU “big data” registries to advance the care and science of NBS. As the cohort matures and grows and more health outcomes are collected, so does the potential for novel insights and new knowledge discovery, advancing the understanding of and improvement in care for patients with NBS conditions.

### 4.2. Mitigating Challenges

The Network employs a comprehensive model to address critical challenges in NBS LTFU, focusing on three crucial areas. Firstly, faced with an expanding NBS population and insufficient public funding, the model utilizes *Epic*’s registries to automate data collection, mitigating resource needs and reducing errors associated with manual data entry. Secondly, integrating registries, health maintenance tools, and dashboards directly into the EHR system ensures real-time data access, enabling timely identification and closure of care gaps for more efficient condition management. Lastly, the model tackles the issue of linking multiple data sources through the registries’ ability to pull multisite data from health information exchanges. This approach addresses interoperability challenges and provides a consolidated perspective on a patient’s clinical, environmental, familial, and socio-economic factors, ultimately optimizing NBS LTFU processes and improving overall patient care outcomes.

### 4.3. Quantitative Evaluation

The Network’s implementation of its LTFU model in the early childhood cohort versus the legacy cohort demonstrates significant and likely sustainable improvements in the %UTD on visits. Promising trends in %UTD on visits in the early childhood cohort and additional condition-specific KPMs over a two-year project period will likely reach significance as the cohort grows. The cohort size displayed fluctuations over time due to various factors, including the addition of new patients diagnosed with specific conditions, individuals relocating out of the state, or those with conditions that were found to be transient. Beyond just assessing LTFU outcomes, the feeding forward of measurable improvements in care delivery to specialty care teams throughout the journey was highly motivating to them.

### 4.4. Qualitative Evaluation

The qualitative evaluation enabled the Network to delve deeply into the specialty care team’s experience of implementing the LTFU model. The feedback gathered through the focus group discussions provided insight into work culture shifts that helped the teams incorporate population health tools into daily practice. Specialty care team engagement was evidenced by their desire to sustain these practice transformations and expand upon them by leveraging registry tools for quality improvement initiatives and replicating the model with patients with other complex chronic conditions not under the NBS umbrella. After facing initial resistance, the Network realized the importance of meeting each care team where they were and customizing dashboard reviews for each department based on their staff and workflows. Over time, the specialty care teams came to appreciate how they could proactively use technology to identify high-risk cases. The Network believes their strong attention to workflow, nurse analyst training, ongoing support, and optimization contributed to the positive care trends. Specialty care teams regarded this practice change as a valued transformation rather than an unwanted disruption. In addition, all specialty care teams expressed a desire to continue using dashboard review for the NBS population and to expand it to other non-NBS populations.

### 4.5. Limitations

In Connecticut, two pediatric treatment centers serve the NBS population, both of which use *Epic*. This use case, which has already shown promising results, was made possible because the Network is embedded within one of the two treatment centers and directly exchanges data with the other through health information exchange supported by a data-sharing agreement. This means that the Network can monitor patients regardless of where they receive care. Plans are underway to expand the model to all of the remaining care teams at both centers.

While this use case may not be fully replicable by most NBS programs, it offers valuable insights and principles that can be adapted. For instance, programs could explore partnerships with specialty care centers to develop comparable registries, enabling consistent data collection and the sharing of aggregated data with relevant health authorities. Moreover, if a state’s health information exchange infrastructure allows for it, health departments could build reports and dashboards pulling data directly from the exchange, enhancing access to LTFU data and improving care coordination efficiency. By leveraging emerging health information technology, programs can enhance their ability to monitor patient outcomes and ensure comprehensive long-term care for individuals identified through NBS.

Another limitation to consider is that this model requires a dedicated analyst trained in *Epic* or other EHR systems, a position not routinely available in many programs. Additionally, the significant effort required to change the practice behaviors of the clinical care teams takes time and resources. However, it is essential to note that even with the adoption of these population health tools, if there is a shortage of providers and support staff, the quality of care for these patients will not see improvement. Therefore, it is imperative that the workforce is adequate to serve the NBS population.

## 5. Conclusions

The Network has successfully spearheaded a transformation of Connecticut’s NBS system by aligning emerging technology and evidence-based practices with the voiced priorities of providers and patients/families. The Network’s LTFU model serves as a use case for applying and achieving the adoption of population health tools within an EHR system to track care delivery and quickly fill identified care gaps, thus improving long-term health for NBS patients.

The ongoing expansion of the LTFU registry to all NBS specialty care teams is underway, with qualitative results shaping continuous program improvements, and the Network is already leveraging the registry to track additional pilot projects looking at social determinants of health and transitioning from adolescence to adulthood.

With the robust registry framework, the Network is well positioned to incorporate more evidence-based tools and patients over time, ensuring a dynamic and responsive approach to improving LTFU health outcomes. By leveraging these strategies and evidence-based interventions, the Network’s goal is to effectively make data-driven improvements in LTFU while making a compelling case to funders of the program’s cost-effectiveness.

## Figures and Tables

**Figure 1 IJNS-10-00027-f001:**
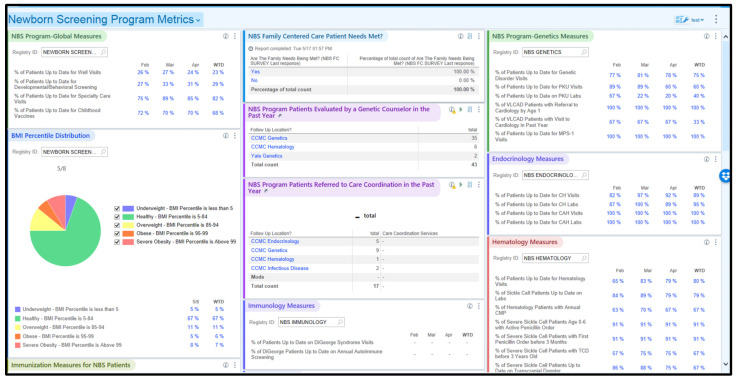
Program-level dashboard displaying KPMs for the entire NBS Network LTFU cohort.

**Figure 2 IJNS-10-00027-f002:**
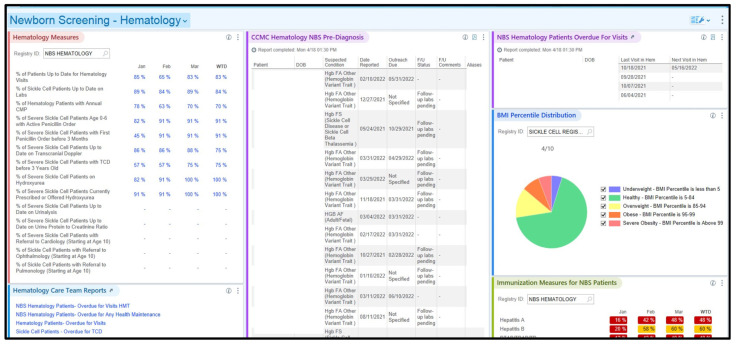
A partial view of a specialty care team dashboard (hematology) that shows KPMs and relevant reports for this specialty. Patient name and date of birth are visible for the specialty care teams but redacted for this visual.

**Figure 3 IJNS-10-00027-f003:**
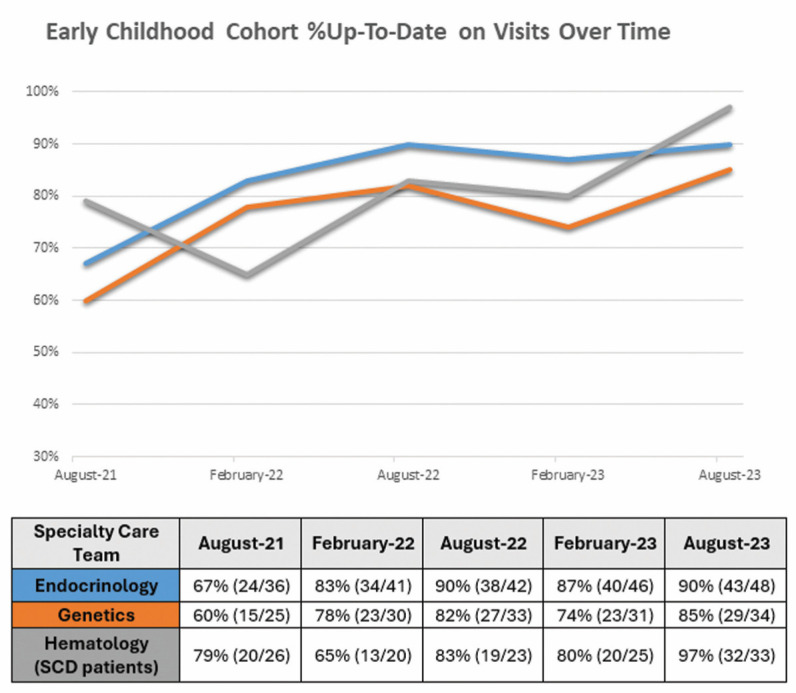
Graph showing early childhood cohort %UTD on visits by specialty care team over time.

**Figure 4 IJNS-10-00027-f004:**
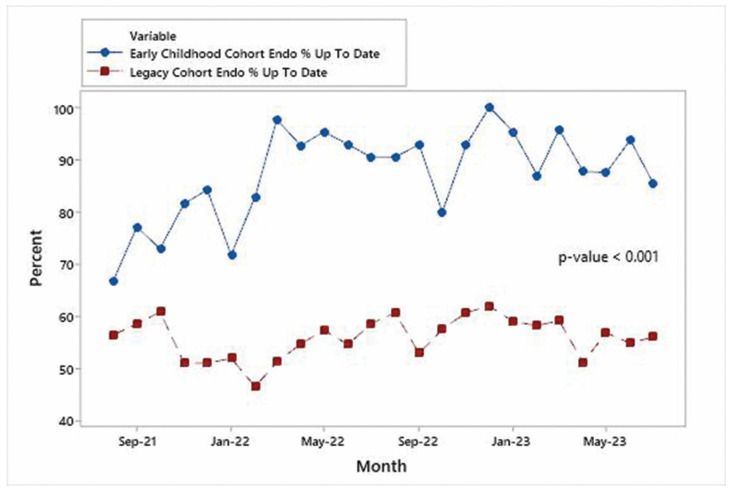
Early childhood endocrinology vs. legacy endocrinology.

**Figure 5 IJNS-10-00027-f005:**
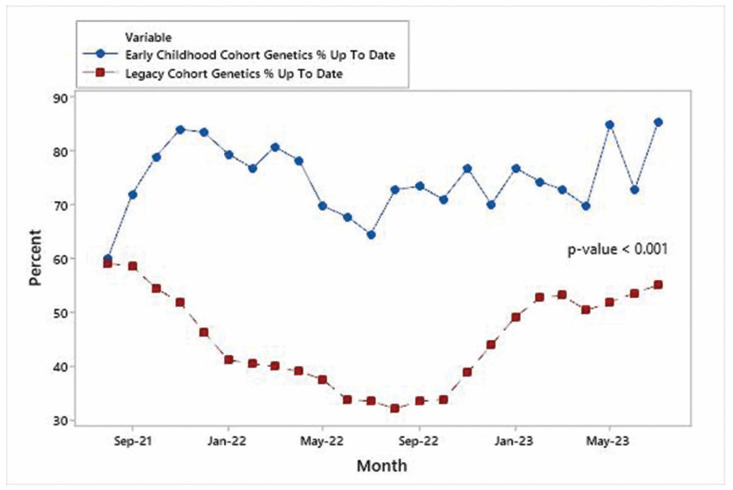
Early childhood genetics vs. legacy genetics.

**Figure 6 IJNS-10-00027-f006:**
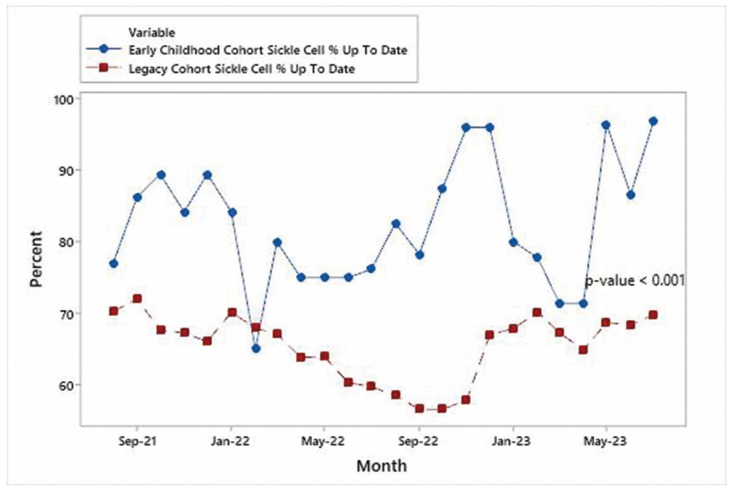
Early childhood hematology SCD vs. legacy hematology SCD patients.

**Figure 7 IJNS-10-00027-f007:**
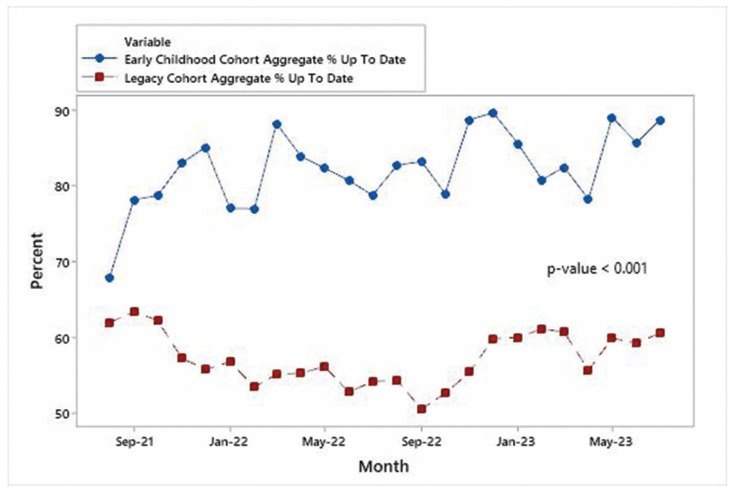
Early childhood cohort vs. legacy cohort aggregate.

**Figure 8 IJNS-10-00027-f008:**
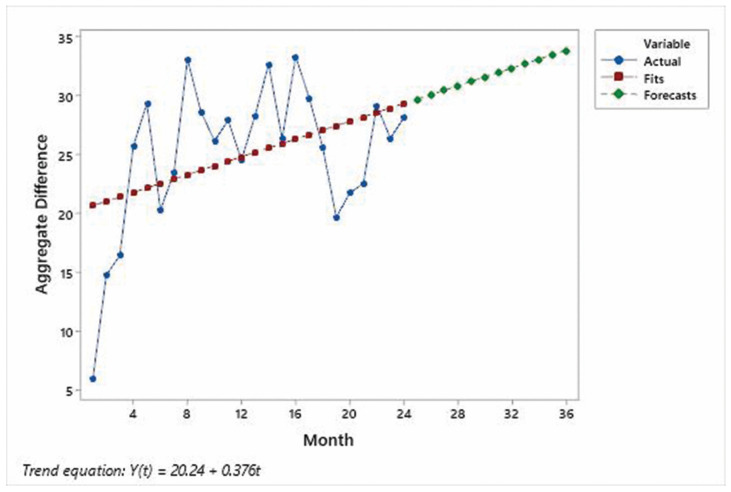
Trend analysis plot for early childhood cohort aggregate difference.

**Table 1 IJNS-10-00027-t001:** Trends in KPMs during project period.

Specialty/ Population	Metric Evaluated ^1^	Aug ‘21 (Baseline)	Feb ‘22	Aug ‘22	Feb ‘23	Aug ‘23
Endocrinology/ Early Childhood Cohort with CH	%UTD on TSH and Free T4 labs	82% (27/33)	100% (38/38)	100% (39/39)	91% (40/44)	95% (42/44)
Genetics/ Early Childhood Cohort with VLCADD	% with cardiology referral in place by age 1 yr	50% (1/2)	100% (3/3)	100% (3/3)	100% (2/2)	100% (2/2)
Hematology/ Early Childhood Cohort with severe SCD aged 2 yr–4 yr	%UTD on TCD ^2^	43% (3/7)	75% (6/8)	90% (9/10)	73% (8/11)	92% (12/13)
Hematology/ Early Childhood Cohort with severe SCD	% currently prescribed or offered hydroxyurea	80% (8/10)	100% (11/11)	100% (11/11)	85% (11/13)	100% (13/13)
Hematology/ Early Childhood Cohort with severe SCD	% with first penicillin order before age 3 mo	50% (5/10)	91% (10/11)	91% (10/11)	85% (11/13)	100% (13/13)

^1^ Note: these are a sample of KPMs prioritized for tracking and reviewing by specialty care teams. ^2^ TCD = transcranial doppler.

## Data Availability

The original contributions presented in the study are included in the article, further inquiries can be directed to the corresponding author.

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
