# Peer review of "Advancing Newborn Screening Long-Term Follow-Up: Integration of Epic-Based Registries, Dashboards, and Efficient Workflows"

_2409-515X, 2024, doi:10.3390/ijns10020027_

Round 1

Reviewer 1 Report

Comments and Suggestions for Authors

This is a very interesting manuscript that provides an example of the use of existing hospital electronic health records to collect short- and long-term follow-up (LTFU) data on babies having a positive newborn screen. The manuscript describes the use of the Connecticut’s Network’s LTFU model and evaluated innovations and emerging technology not previously described in the literature, which provided methods to effectively reduce barriers to collection of LTFU data.  The system that was used EPICs “Care Everywhere” allowed for retrieving data from primary care practices, imaging facilities, and hospitals with no requirement for extra manual data entry. The interoperability of the associated systems resulted in a more unified representation of the patient’s long-term care journey and allowed for improving the care of the affected individuals through use of dashboards showing key performance metrics. 

There was a great amount of effort put into assembling the appropriate teams for each condition and setting up the system and dashboards for the chosen disorders.  The only thing that might improve the paper is a little discussion on the general applicability of this approach to other programs.

Reviewer 2 Report

Comments and Suggestions for Authors

The authors make a good description of how Epic can support LTFU in NBS settings. This is an important demonstration as follow-up is a challenge given the importance of screening in diagnosis and care and given this is an important approach as NBS can expand. The main issue noted here is it is difficult to understand still how this is helping and how it will be useful across other NBS disorders or where the limits are of this approach. What improvements and Epic modules are needed? And while one can observe data who is responsible. Some of this and the transitions are not clear for the audience to understand. What will happen if a child has a change in insurance or moves out of state? Some brief discussions will help.

A few minor issues noted below:

a) Collaboration /w New Haven Hospital but not the entire states NBS. Can there be a comment on how this will be expanded? It is unclear.

b) Line 94… What are the 200 metrics?

c) Line 98 Why is cohort growth 60-70 a year? Is this at max possible?

d) Line 256 Fig 4 -8. Y axis should read e.g 30% and not 0.3
